# Systematic Review and Meta-Analysis of the Efficacy and Effectiveness of Pneumococcal Vaccines in Adults

**DOI:** 10.3390/pathogens12050732

**Published:** 2023-05-19

**Authors:** Jennifer L. Farrar, Lana Childs, Mahamoudou Ouattara, Fahmina Akhter, Amadea Britton, Tamara Pilishvili, Miwako Kobayashi

**Affiliations:** 1Respiratory Diseases Branch, U.S. Centers for Disease Control and Prevention, Atlanta, GA 30329, USA; 2CDC Foundation, Atlanta, GA 30308, USA

**Keywords:** pneumococcal vaccine, invasive pneumococcal disease, pneumonia, adults, vaccine effectiveness

## Abstract

New pneumococcal conjugate vaccines (PCVs), 15- and 20-valent (PCV15 and PCV20), have been licensed for use among U.S. adults based on safety and immunogenicity data compared with the previously recommended 13-valent PCV (PCV13) and 23-valent pneumococcal polysaccharide vaccines (PPSV23). We conducted a systematic review of the literature on PCV13 and PPSV23 efficacy (randomized controlled trials [RCTs]) or effectiveness (observational studies) against vaccine type (PCV13 type or PPSV23 type, respectively), invasive pneumococcal disease (IPD), and pneumococcal pneumonia (PP) in adults. We utilized the search strategy from a previous systematic review of the literature published during the period from January 2016 to April 2019, and updated the search through March 2022. The certainty of evidence was assessed using the Cochrane risk-of-bias 2.0 tool and the Newcastle–Ottawa scale. When feasible, meta-analyses were conducted. Of the 5085 titles identified, 19 studies were included. One RCT reported PCV13 efficacy of 75% (PCV13-type IPD) and 45% (PCV13-type PP). Three studies each reported PCV13 effectiveness against PCV13-type IPD (range 47% to 68%) and against PCV13-type PP (range 38% to 68%). The pooled PPSV23 effectiveness was 45% (95% CI: 37%, 51%) against PPSV23-type IPD (nine studies) and 18% (95% CI: −4%, 35%) against PPSV23-type PP (five studies). Despite the heterogeneity across studies, our findings suggest that PCV13 and PPSV23 protect against VT-IPD and VT-PP in adults.

## 1. Introduction

The bacteria *Streptococcus pneumoniae* are a common cause of bacterial pneumonia, bacteremia, and meningitis in adults, especially among older adults and those with certain underlying conditions that increase the risk of pneumococcal disease. According to the U.S. Centers for Disease Control and Prevention (CDC)’s Active Bacterial Core surveillance, invasive pneumococcal disease (IPD) incidence among U.S. adults aged ≥ 65 years was 24 per 100,000 during 2018–2019, and disease caused by serotype 3, a serotype included in the previously recommended 13-valent pneumococcal conjugate vaccine (PCV13) and 23-valent pneumococcal polysaccharide vaccine (PPSV23), is the most common vaccine serotype causing IPD in adults, with limited evidence of change in incidence despite vaccine use [1,2,3,4,5]. Incidence rates for pneumococcal pneumonia among adults aged ≥ 65 years ranged from 33 to 100 per 100,000 between 2010 and 2017 [6,7,8]. In a 2016 study of U.S. adults hospitalized with radiologically-confirmed community-acquired pneumonia, 8.2% (*n* = 520) of adults ≥ 65 years had *S. pneumoniae* detected by urine antigen, and of these, 269 (51.7%) were serotypes included in PCV13 [5].

Until recently, the U.S. Advisory Committee on Immunization Practices (ACIP) recommended two pneumococcal vaccines, PCV13 and PPSV23, for use in adults to prevent disease caused by *S. pneumoniae*. Since 2012, PCV13 has been recommended for use in adults aged ≥ 19 years with an immunocompromising condition, cochlear implant, or cerebrospinal fluid leak. PPSV23 has been in use since 1984, and, until recently, was recommended for all adults aged ≥ 65 years and adults aged ≥ 19 years with underlying medical conditions or other risk factors that increase the risk of pneumococcal disease [9]. While studies have shown that PPSV23 is effective against PPSV23-type IPD, data on the effectiveness of PPSV23 against nonbacteremic pneumococcal pneumonia have been inconsistent [10]. In contrast, a large randomized controlled trial (RCT) in adults aged ≥ 65 years in the Netherlands showed that PCV13 is effective against PCV13-type IPD and PCV13-type nonbacteremic pneumococcal pneumonia [11]. Data from this study supported the 2014 recommendation for routine PCV13 use for all adults aged ≥ 65 years in the United States [10]; however, additional years of data showed that there was a limited population-level impact of routine PCV13 use in adults of this age group, likely due to the already realized indirect effects from PCV13 use in children. As a result, in 2019, PCV13 was no longer routinely recommended for adults aged ≥ 65 years without immunocompromising conditions, cerebrospinal fluid leaks, or cochlear implants, and instead was recommended based on shared clinical decision-making [12]. In 2021, two new pneumococcal conjugate vaccines (PCVs) were licensed for use in the United States: 15-valent PCV (PCV15) and 20-valent PCV (PCV20) [13,14].

A review of PCV13 and PPSV23 effectiveness against clinical outcomes was necessary for a few reasons: First, given that PCV15 and PCV20 were licensed based on safety and immunogenicity data only, this review was expected to help inform the estimated effectiveness of PCV15 and PCV20 against clinical outcomes in adults. Second, PPSV23 was likely to be part of the new recommendations, and the previous literature found inconsistent evidence regarding PPSV23 effectiveness against nonbacteremic pneumococcal pneumonia. Therefore, we conducted a systematic review of the literature on the efficacy or effectiveness of PCV13 and PPSV23 against respective vaccine type (VT)-IPD and VT-pneumococcal pneumonia in adults to help inform the U.S. ACIP’s discussions on the use of PCV15 and PCV20 in adults.

## 2. Materials and Methods

### 2.1. Literature Search

We conducted a systematic literature search reviewing seven databases [Medline, Embase, CINAHL, Web of Science, Scopus (Web of Science), Epistemonikos, and Cochrane Library] for relevant data published from April 2019 to February 2021, employing a search strategy (Appendix A) utilized by the Norwegian Public Health Institute (NIPH) to evaluate pneumococcal vaccine effectiveness in adults for the World Health Organization’s Strategic Advisory Group of Experts discussion [15]. All relevant studies from the previous search by NIPH (published from January 2016 to March 2019) were also included in our review. Authors were contacted for additional information whenever possible. The results of all searches were entered into the Covidence systematic review software program [16].

The protocol for this review was developed in line with PRISMA-P recommendations [17] and registered with PROSPERO (CRD42021258668). The protocol was amended in February 2022 to include an additional year of searching that was performed after the initial review. This search was conducted on 4 April 2022 to identify additional titles published from 13 February 2021 to 15 March 2022 to update the review.

### 2.2. Inclusion and Exclusion Criteria

Data on PCV13 and PPSV23 efficacy or effectiveness against IPD (defined as detection of *S. pneumoniae* in normally sterile sites, such as bacteremia and meningitis) and pneumococcal pneumonia (defined as pneumonia due to a *S. pneumoniae*) caused by serotypes included in the evaluated vaccines were included. For the purposes of this review, vaccine-type (VT) refers to the serotypes included in the vaccine evaluated (e.g., if the study evaluated PCV13 against VT-IPD, VT for the study includes the 13 serotypes included in PCV13 unless otherwise noted).

We included published RCTs and observational studies (prospective and retrospective comparative cohort studies, case-control or nested case-control studies, indirect cohort studies, screening method studies) from middle- and high-income settings that evaluated the direct effects of pneumococcal vaccination on adults (≥16 years). Systematic reviews on PCV13 or PPSV23 efficacy or effectiveness were included to leverage their bibliographies for relevant data. Only English language studies were included.

We excluded studies assessing vaccine impact, case reports, case-series, animal studies, modeling studies, health economic evaluations, pneumococcal carriage studies, narrative reviews, studies that only included data on vaccines that are not currently used in the United States (e.g., PCV7, PCV10), studies evaluating the indirect effects of pediatric pneumococcal vaccination on adult populations, and studies specifically targeting adults with immunocompromising conditions. In addition, because pneumococcal disease epidemiology differs from that in the United States, we excluded studies conducted in low-income settings.

### 2.3. Data Abstraction and Cleaning

Titles and abstracts of all citations identified through the literature search were independently screened for relevance and inclusion by two reviewers using Covidence [16]. The articles meeting inclusion criteria following title and abstract screening were then double screened through a full-text review. Discordant reviews during title and abstract or full-text review were blindly reviewed for inclusion by an independent third reviewer.

Data were independently abstracted by two reviewers for study information (study characteristics, methods, results) using standardized data collection tools and entered into a Microsoft Excel spreadsheet.

Data were also assessed for certainty of evidence. Randomized studies were assessed for certainty using the Cochrane risk-of-bias 2.0 tool [18], which included consideration of the appropriate generation of random allocation sequences; concealment of the allocation sequences; blinding of participants, healthcare providers, data collectors, and outcome adjudicators; and proportion of patients lost to follow-up. Non-randomized studies were assessed considering the data elements included in the Newcastle–Ottawa scale [19] to address potential sources of bias in cohort and case-control studies. Each study was assessed for risk of bias independently by two reviewers. A third reviewer was consulted when there was discordance of the overall certainty of evidence between the two independent reviewers. Studies with scores ≥ 7 (out of 9) were deemed high quality/low risk of bias, scores between 4 and 6 (out of 9) were deemed medium quality/some risk of bias, and scores ≤ 3 were deemed low quality/high risk of bias.

### 2.4. Data Analysis

We conducted descriptive analyses of studies stratified by outcome (PCV13-type IPD, PPSV23-type IPD, PCV13-type pneumococcal pneumonia, or PPSV23-type pneumococcal pneumonia), vaccine product (PCV13 or PPSV23), and study design (RCT or observational study). When feasible, we conducted meta-analyses by outcome, product, and study design using Comprehensive Meta-Analysis Version 3 [20]. Random effects models were used to pool estimates by stratum. Because PPSV23 has been used in adults for much longer than PCV13, the time since PPSV23 vaccination was more variable across studies compared with PCV13 studies. To account for the waning of vaccine effectiveness over time [21,22,23,24,25,26], we limited the PPSV23 analysis to data from individuals who were vaccinated within 5 years of assessment. When data were available, we performed sub-analyses by underlying conditions and limiting the sample to a younger age group of adults aged from 65 to 74 years to account for heterogeneity in the study population between observational studies.

## 3. Results

### 3.1. Literature Search

The literature search yielded a total of 5085 articles, 2499 from the NIPH search that covered titles published from January 2016 to April 2019, and 2586 from the updated search that covered titles published from April 2019 to March 2022. From the updated search, 1764 studies were screened for eligibility; 1525 studies were excluded during title and abstract screening. During the full-text review, 239 articles were screened for eligibility and 229 were excluded. Ten studies were included from the updated searches and nine studies were included from the NIPH search, totaling nineteen studies for inclusion in the analysis (Figure 1). Seven studies evaluated PCV13 and thirteen evaluated PPSV23; one study evaluated both PCV13 and PPSV23 against PCV13 type- and PPSV23 type-pneumococcal pneumonia [27]. Efficacy or effectiveness against IPD caused by VT-serotypes was evaluated in thirteen studies, and against pneumococcal pneumonia caused by serotypes included in the vaccine was evaluated in eight studies; two studies evaluated pneumococcal vaccine effectiveness against both VT-IPD and VT-pneumococcal pneumonia (Table 1) [11,28].

### 3.2. Vaccine-Type Invasive Pneumococcal Disease

#### 3.2.1. PCV13

We identified four studies evaluating PCV13 efficacy or effectiveness against PCV13-type IPD. Only one RCT (CAPITA) was identified, with a reported efficacy of 75% (95% CI: 41%, 91%) against PCV13-type IPD in Dutch pneumococcal vaccine-naïve, community-dwelling adults aged ≥65 years [11]. Vaccine effectiveness (VE) estimates from three observational studies ranged from 47% to 68% against PCV13-type IPD in adults aged ≥65 years [29,31,32] (Table 1). Lewis et al. evaluated PCV13 VE in a cohort of U.S. adults aged ≥65 years who were enrolled in Kaiser Permanente Northern California insurance and had not received PPSV23 before the age of 65 years. PCV13 VE was 68% (95% CI: 38%, 84%) against all PCV13 IPD, and 53% (95% CI: −10%, 80%) against serotype 3 IPD; the study reported an increase in the incidence of IPD during the study period (2014–2018), largely due to serotype 3 [29]. The other two observational studies by Pilishvili et al. were case-control studies that used Active Bacterial Core surveillance data to identify cases, but different sources to identify controls [31,32]. The VE estimate for PCV13 against IPD caused by PCV13 + 6C serotypes was 59% (95% CI: 11%, 81%), using controls identified from the commercial database ReferenceUSAGov (InfoGroup) [31] and 47% (95% CI: 4%, 71%) using Medicare beneficiaries as controls; this estimate increased to 67% (95% CI: 11%, 88%) when excluding serotype 3 IPD cases, which was the most common serotype causing disease [32]. Due to overlap in cases between the two Pilishvili et al. studies, a pooled VE was not estimated between studies.

#### 3.2.2. PPSV23

We did not identify any RCTs assessing PPSV23 efficacy against PPSV23-type IPD. Nine observational studies reported VE among those who received PPSV23 within 5 years of the study assessment [21,23,24,25,28,35,37,38,39]. Of these, seven were indirect cohorts and two were case-control studies. The studies evaluated VE among adults aged ≥60 years from Canada, Germany, the United Kingdom (*n* = 2), South Korea, Japan, Taiwan, and Spain (*n* = 2). PPSV23 VE estimates for these studies ranged from 42% to 64% for PPSV23-type IPD. Two studies also evaluated PPSV23 effectiveness against PCV13-type IPD: Kim et al. reported −15% (95% CI: −122%, 41%) effectiveness among adults aged ≥ 65 years and Su et al. reported 35% and 40% effectiveness for PCV13 minus 6A-type IPD using the indirect cohort and screening methods, respectively [28,39]. Serotype 3 IPD estimates ranged from −110% to 37%, although most were not statistically significant [21,23,25,28,37,38,39]. Two studies, Kim et al. and Perniciaro et al., reported improved PPSV23 VE against PPSV23-type IPD when serotype 3 disease was excluded from the analysis [28,37]. In addition, Perniciaro et al. reported a −29% (95% CI: −159%, 31%) PPSV23 effectiveness against PPSV23-type IPD among those vaccinated <2 years; the authors observed that over 36% of cases were caused by serotype 3. Therefore, we report the VE against PPSV23/non-PCV13 serotype IPD, which was 35% (95% CI: −14%, 65%) [37]. The pooled VE estimate from these nine observational studies was 45% (95% CI: 37%, 51%; I^2^ = 0%) (Figure 2). When limited to three studies in adults without immunocompromising conditions [21,23,24], the pooled VE estimate was 60% (95% CI: 47%, 69%; I^2^ = 0%) (Figure 3). The pooled VE estimate for a sub-analysis restricted to adults aged from 65 to 74 years [23,24,28] was 52% (95% CI: 36%, 64%; I2 = 5.2%) (Figure 4).

### 3.3. Vaccine-Type Pneumococcal Pneumonia

#### 3.3.1. PCV13

We identified four studies evaluating PCV13 efficacy or effectiveness against PCV13-type pneumococcal pneumonia. One RCT, among Dutch adults aged ≥ 65 years, reported an efficacy of 45% (95% CI: 14%, 65%) against the first episode of PCV13-type nonbacteremic and noninvasive pneumococcal pneumonia [11]. In addition to the RCT, we identified three test-negative design (TND) observational studies for PCV13 [27,30,33]. The study by Prato et al. was conducted among outpatient and inpatient Italian adults aged ≥ 65 years living in the Apulia region [33]. Many participants had one or more underlying comorbidity including chronic heart disease (53%), chronic respiratory disease (44%), and diabetes (25%). Patients were considered to have pneumococcal pneumonia if they had a positive PCR result for *S. pneumoniae* from blood, sputum, or bronchoalveolar-lavage. Prato et al. reported a crude VE of 38% (95% CI: −132%, 89%) against PCV13-type pneumococcal pneumonia, and an adjusted VE was not reported [33]. The study by McLaughlin et al. was conducted as a nested study within a larger population-based surveillance study of U.S. adults in Kentucky. Adults aged ≥65 years and hospitalized with community acquired pneumonia (CAP) were considered for enrollment. Most participants (88%) had one or more at-risk (defined as immunocompetent, but presence of a chronic disease such as congestive heart failure, diabetes mellitus, or liver disease) or high-risk (defined as immunocompromised) condition, and almost half of participants were considered high-risk (46%). The adjusted VE was 68% (95% CI: −6%, 90%) against nonbacteremic PCV13-type pneumococcal pneumonia [30]. Heo et al. evaluated the effectiveness of PCV13, PPSV23, and sequential PCV13/PPSV23 vaccination against pneumococcal pneumonia among older South Korean adults aged ≥65 years hospitalized with CAP. The mean age of participants was 76.7 ± 6.9 years and many participants had ≥1 underlying comorbidity (80%) or ≥2 underlying comorbidities (45%). The adjusted VE against PCV13-type pneumococcal pneumonia was 41% (95% CI: −104%, 83%) for all adults aged ≥ 65 years [27]. Pooled VE of the TND observational studies was not estimated since Prato et al. only reported a crude VE estimate.

#### 3.3.2. PPSV23

We did not identify any RCTs evaluating the efficacy of PPSV23 against PPSV23-type pneumococcal pneumonia. We identified five observational studies with VE estimates ranging from −2% to 46% [22,27,28,34,36]; three studies also reported PPSV23 estimates against PCV13-type pneumococcal pneumonia with estimates ranging from −24% to 40% [22,27,28]. The study by Kim et al. was a hospital-based case-control study conducted among South Korean adults aged ≥ 65 years. The median age of the study population was 76 years and many participants had one or more underlying medical conditions such as chronic pulmonary disease (37% of cases, 23% of controls) or diabetes mellitus (24% of cases, 30% of controls). Approximately 26% of cases and 37% of controls had an immunocompromising condition. The adjusted VE against PPSV23-type nonbacteremic pneumococcal pneumonia was −2% (95% CI: −40%, 26%). The authors reported that 14% of cases were caused by serotype 3; the vaccine effectiveness estimate for PPSV23-type pneumococcal pneumonia excluding serotype 3 cases was not protective for any age group [28]. The four remaining observational studies were TND. Lawrence et al. used a nested TND of a prospective cohort study of adults aged ≥16 years hospitalized with CAP at two hospitals in England. The mean age was similar between cases and controls (66.5 versus 65.4 years). The most common comorbidities among cases and controls were chronic obstructive pulmonary disease (COPD) (24% for cases and controls), chronic lung disease (27% for cases and controls), and hypertension (24% of cases, 25% of controls). The adjusted VE of PPSV23 against PPSV23-type pneumococcal pneumonia (non-bacteremic and bacteremic) among those vaccinated <5 years was –7% (95% CI: −54%, 26%). Lawrence et al. observed that serotype 5 caused the largest proportion of cases of PPSV23-serotype pneumonia <5 years since vaccination, and considered the potential cross-reactivity with other streptococcal strains that express serotype 5; however, the authors noted the potential cross-reactivity did not alone explain the difference in effects among vaccinated and unvaccinated patients. Therefore, we report the VE against PPSV23/non-PCV13 serotypes pneumococcal pneumonia, which was 46% (95% CI: 5%, 69%). Lawrence et al. also evaluated PPSV23 effectiveness against serotype 3 pneumococcal pneumonia with an adjusted estimate of 40% (95% CI: 14%, 59%) [36]. Suzuki et al. conducted a prospective TND study among adults aged ≥65 years at four community-based hospitals in Japan. The presence of underlying medical conditions was similar between cases and controls. The adjusted VE of PPSV23 against PPSV23-type pneumococcal pneumonia among outpatients and inpatients was 34% (95% CI: 6%, 53%) and 41% (95% CI: −11%, 69%) against serotype 3 pneumococcal pneumonia [22]. Heo et al., described above, reported an adjusted VE of 6% (95% CI: −74%, 50%) against PPSV23-type pneumococcal pneumonia among participants aged ≥65 years [27]. Chandler et al. conducted a TND study among hospitalized CAP patients identified through the University of Louisville Pneumonia Study database, which was also used for the study by McLaughlin et al. [30,34]. The median age of vaccinated patients was 68 years compared to 65 years among non-vaccinated patients. Vaccinated patients were more likely to have comorbid conditions such as COPD (58% among vaccinated patients versus 47% among unvaccinated patients), renal failure (33% versus 26%), or diabetes (37% versus 31%). Chandler et al. reported 2% (95% CI: −50%, 38%) vaccine effectiveness against PPSV23-type pneumococcal pneumonia in adults aged ≥65 years who received PPSV23 within 5 years of assessment [34]. The pooled VE of PPSV23 against PPSV23-type pneumococcal pneumonia from the five observational studies was 18% (95% CI: −4%, 35%; I^2^ = 0%) (Figure 5). When limited to adults aged from 65 to 74 years, the pooled VE estimate from three studies [22,27,28] was 26% (95% CI: −6%, 49%; I^2^ = 0%) (Figure 6).

### 3.4. Certainty of Evidence

The risk of bias for the one RCT [11] was low; therefore, the certainty of evidence was high for this study. Regarding observational studies, scores using the Newcastle–Ottawa scale ranged from 3 to 8 (out of 9) for PCV13 studies; five studies were deemed high quality (score of 7 or 8) [27,29,30,31,32]. Prato et al. scored 3 for lack of information regarding the selection of controls and exposure criteria, as well as a lack of controlling for potential confounders; it was rated low for quality of evidence [33]. For PPSV23 observational studies, eight studies were deemed high quality (score of 7 to 8) [22,24,25,27,28,34,37,39] and five were deemed medium quality (score of 4 to 6) [21,23,35,36,38]. All studies that scored medium quality lacked information regarding the source of controls or the ascertainment of exposure (i.e., how vaccination history was obtained).

## 4. Discussion

Recent data continue to support that both PCV13 and PPSV23 are effective against VT-IPD in adults. However, there are a few differences in the published studies that warrant additional discussion. When considering PCV13, the VE estimates from Pilishvili et al. were lower than the estimates from the RCT; this is expected, as effectiveness estimates from observational studies are often lower than efficacy estimates from clinical trials, which occur in a more controlled setting. Furthermore, the lower observed estimates could be due to differences in the study population. Specifically, 54–60% of cases and 31–32% of controls in the case-control studies by Pilishvili et al. had immunocompromising conditions, whereas adults with immunocompromising conditions at enrollment were not included in the RCT. Regarding PCV13 effectiveness against PCV13-type pneumococcal pneumonia, findings from observational studies supported findings from the RCT, which showed 45% efficacy. However, estimates from the observational studies had wide confidence intervals.

When considering PPSV23, the pooled VE estimate for PPSV23 against PPSV23-type IPD was slightly lower than what was observed for PCV13, even after limiting the data to <5 years since vaccination. These findings are not unexpected as immunogenicity studies have observed polysaccharide vaccines to be less immunogenic compared to conjugate vaccines. This review reinforced the existing evidence for PPSV23 effectiveness against PPSV23-type IPD.

Until recently, there were limited data specifically assessing PPSV23 VE against PPSV23-type pneumococcal pneumonia. Previous studies that reported PPSV23 effectiveness against pneumococcal pneumonia or CAP had variable results [40,41,42]. A review and meta-analysis by Falkenhorst et al. estimated a pooled VE against pneumococcal pneumonia of 64% (95%CI: 35%, 80%) in clinical trials and 48% (95%CI: 25%, 63%) in cohort studies, after excluding studies with a high risk of bias [40]. Conversely, a review by Schiffner-Rohe et al. reported that three of four studies showed no efficacy against pneumococcal pneumonia [41]. Recent studies assessing PPSV23 effectiveness against PPSV23-type pneumococcal pneumonia allowed us to conduct a pooled VE estimate for this outcome, which suggests that PPSV23 provides some protection against PPSV23-type pneumonia within 5 years of vaccination.

The studies included in this review found that PCV13 offers some protection against serotype 3 IPD. Though Lewis et al. observed a 53% VE against serotype 3 IPD, the findings were not statistically significant and authors also noted an increase in IPD incidence during the study period due to serotype 3 [29]. While PCV13 may provide protection against serotype 3 disease, the vaccine may not mount enough antibody response for sustained indirect effects, and the direct protection from serotype 3 may also be short-lived [43,44]. The effectiveness of PPSV23 against serotype 3 IPD varied and most estimates were not statistically significant. A few studies that evaluated PPSV23 VE, excluding serotype 3, reported higher VE against PPSV23-type IPD [28,37]. There was no evidence for PCV13 effectiveness against serotype 3 pneumococcal pneumonia and limited evidence of some protection from PPSV23 against serotype 3 pneumococcal pneumonia [36]. Despite inclusion in both PCV13 and PPSV23, serotype 3 continues to be the most common vaccine serotype causing disease in U.S. adults [4,5]. As higher valent PCVs become widely available for adult use, evidence regarding vaccine effectiveness against serotype 3 is necessary, given that these vaccines were licensed based on immunogenicity study findings only.

This review supports findings from a similar review conducted earlier by NIPH that reported vaccine effectiveness from PCV13 and PPSV23 against VT- pneumonia, and PPSV23 against PPSV23-type IPD [15]. The inclusion of additional years of data allowed us to conduct a systematic review specifically targeting VE against VT- disease; the previous review by NIPH only included two studies each that assessed VE against PCV13- and PPSV23-type pneumococcal pneumonia, and no study that assessed PCV13 against PCV13-type IPD was included. Our study provides additional evidence that both vaccines are protective against VT- pneumonia, and provides data on PCV13 VE against PCV13-type IPD from observational studies that were conducted after the CAPITA trial. Our review also provides a summary of PCV13 and PPSV23 effectiveness against diseases caused by serotype 3.

There are limitations to our review. First, we identified only one RCT that evaluated PCV13 [11] and none that evaluated PPSV23 efficacy against VT-pneumococcal disease in adults. RCTs provide the most rigorous evidence of vaccine efficacy. However, we identified several observational studies evaluating both PCV13 and PPSV23 effectiveness against VT-IPD and VT-pneumonia. Second, we did limit the outcomes of our review to VT disease as the focus of our review was on the performance of PCV13 and PPSV23 against clinical disease caused by serotypes included in the respective vaccines. However, we acknowledge that non-vaccine type disease (i.e., disease caused by serotypes not included in the vaccines) is an important outcome for public health decision-making and can provide evidence regarding serotype replacement or the cross-protection of related serotypes. Three studies included in this review did provide VE estimates for non-vaccine type disease: Suzuki et al. reported a 2% VE against non-PPSV23 pneumococcal pneumonia [22]. The Pilishvili et al. studies reported 13% and 22% VE estimates against non-PCV13 IPD [31,32]. None of these VE estimates were statistically significant. Third, heterogeneity in study populations, e.g., different age distributions and different proportions of people with underlying conditions included, for PCV13 and PPSV23 studies may have affected the VE estimates. Furthermore, these differences in populations may in part contribute to the range of results observed across studies, though we have attempted to address this limitation by conducting a stratified analysis of the data. Despite these limitations, this was an exhaustive and systematic review of the literature on pneumococcal vaccine products for adult use and provides a comprehensive, current landscape of the evidence regarding VE for PCV13 and PPSV23 against VT-IPD and VT-pneumococcal pneumonia in adults.

In conclusion, our review showed that PCV13 and PPSV23 are both effective against VT-IPD and VT-pneumococcal pneumonia in adults. These data have helped to inform discussions on PCV15 and PCV20 use in adults in the United States, which currently do not have efficacy or effectiveness data against clinical outcomes and thus could be used to inform similar decisions in other countries. Post-licensure effectiveness and impact studies of PCV15 and PCV20 on pneumococcal disease in adults are needed to assess the impact on clinical outcomes.

## Figures and Tables

**Figure 1 pathogens-12-00732-f001:**
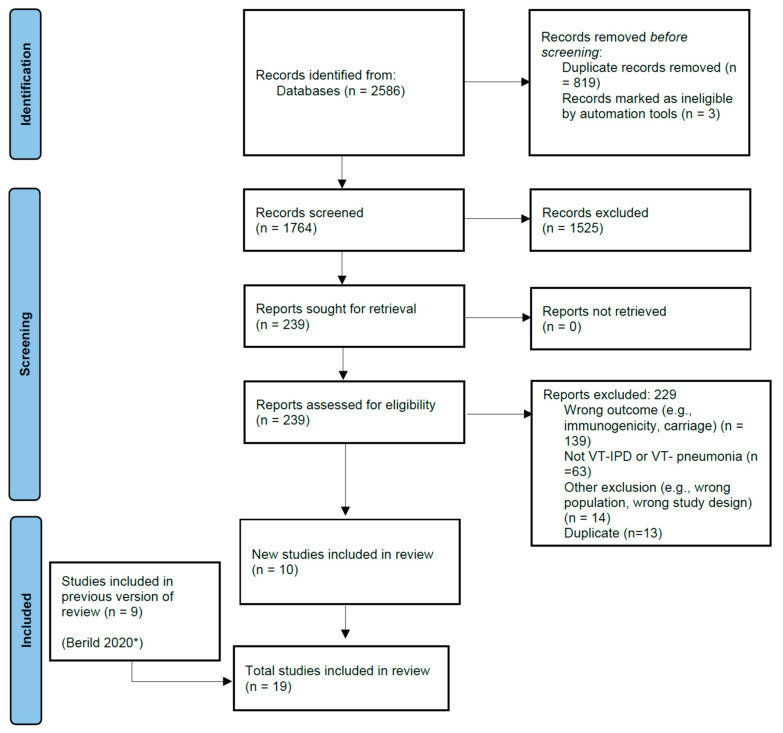
PRISMA Study flow diagram. * Berild 2020 is the publication for the Norwegian Public Health Institute (NIPH) systematic review of literature on pneumococcal vaccine effectiveness in adults conducted for the World Health Organization’s Strategic Advisory Group of Experts.

**Figure 2 pathogens-12-00732-f002:**
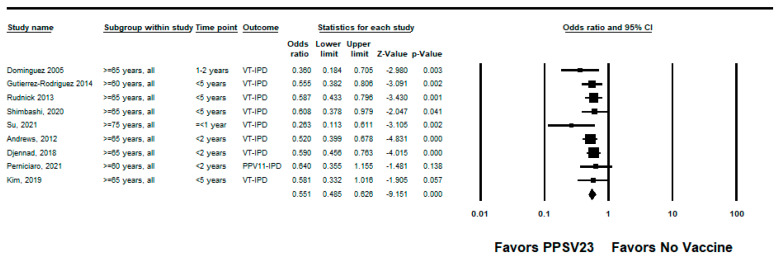
Pooled vaccine effectiveness estimate of 23-valent pneumococcal polysaccharide vaccine against PPSV23-type invasive pneumococcal disease in adults aged ≥60 years: observational studies, <5 years since vaccination [21,23,24,25,28,35,37,38,39].

**Figure 3 pathogens-12-00732-f003:**
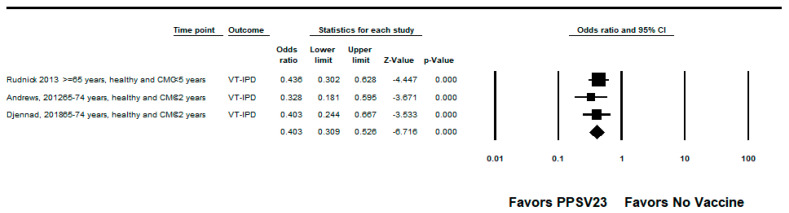
Pooled vaccine effectiveness of 23-valent pneumococcal polysaccharide vaccine against PPSV23-type invasive pneumococcal disease in adults without immunocompromising conditions: observational studies, <5 years since vaccination [21,23,24].

**Figure 4 pathogens-12-00732-f004:**
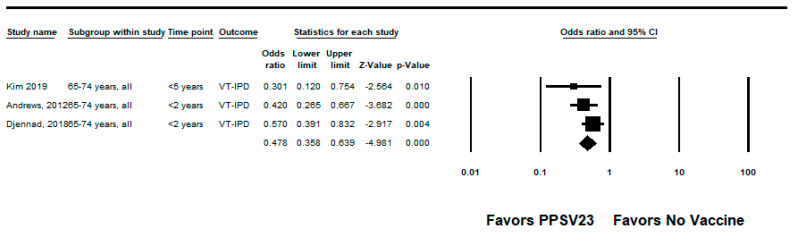
Pooled vaccine effectiveness of 23-valent pneumococcal polysaccharide vaccine against PPSV23-type invasive pneumococcal disease in adults aged 65–74 years: observational studies, <5 years since vaccination [23,24,28].

**Figure 5 pathogens-12-00732-f005:**
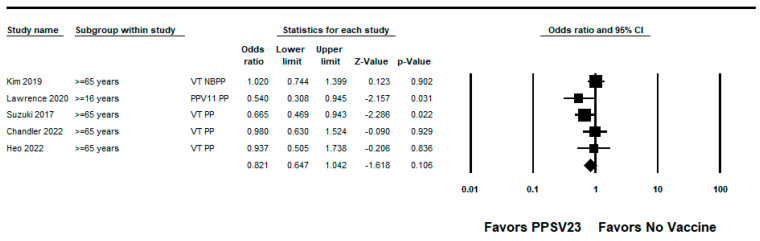
Pooled vaccine effectiveness of 23-valent pneumococcal polysaccharide vaccine against PPSV23-type pneumococcal pneumonia in adults aged ≥16 years: observational studies, <5 years since vaccination [22,27,28,34,36].

**Figure 6 pathogens-12-00732-f006:**
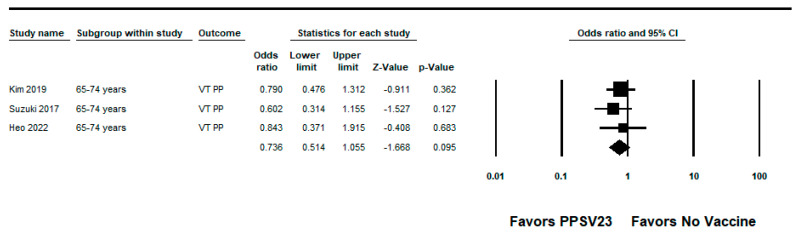
Pooled vaccine effectiveness of 23-valent pneumococcal polysaccharide vaccine against PPSV23-type pneumococcal pneumonia in adults aged 65–74 years: observational studies, <5 years since vaccination [22,27,28].

**Table 1 pathogens-12-00732-t001:** Characteristics of Included Pneumococcal Vaccine Effectiveness Studies.

Author and Publication Year	Country	Population	Study Design	Study Period	Outcome	Factors Adjusted for in VE Estimate, if Reported	VE (95% CI) ^2^	Certainty of Evidence
**PCV13**								
Bonten, 2015 [11] ^1^	The Netherlands	Dutch pneumococcal vaccine-naïve, community-dwelling adults ≥65 years	Randomized Controlled Trial (CAPITA)	2008–2013	PCV13-type IPD	Not applicable	75% (41, 91)	High ^3^
					Non-invasive non-bacteremic PCV13-type CAP	Not applicable	45% (14, 65)	
Heo, 2022 [27]	South Korea	Hospitalized adults ≥65 years	Test-negative	2015–2017	PCV13-type community-acquired pneumonia	Age, sex, risk group based on underlying conditions, disease severity according to the CURB-65 score (confusion, urea level, respiratory rate, blood pressure, age ≥ 65), influenza vaccination status, and 23-valent pneumococcal polysaccharide vaccination status	41% (−104, 83)	High
Lewis, 2019 [29] ^1^	US	Kaiser Permanente Northern California members with no record of prior receipt of PPSV23, adults ≥65 years	Cohort study	2014–2018	PCV13-type IPD	Not reported	68% (38, 84)	High
McLaughlin, 2018 [30] ^1^	US	Inpatient adults ≥65 years	Test-negative	2013–2016	Non-bacteremic PCV13-type pneumococcal pneumonia	Seasonality, age, gender, race, ethnicity, place of residence, risk level, BMI category, pneumonia severity index, healthcare facility exposure in last 3 months, weekly exposure to child <5, influenza vaccination in previous year, PPSV23 vaccination within 5 years	68% (−6, 90)	High
Pilishvili, 2018 [31]	US	Active Bacterial Core Surveillance (ABCs) IPD cases, adults ≥65 years	Case-control; controls identified using ReferenceUSAGov	2015–2017	PCV13-type + 6C IPD	Presence of chronic and immunocompromising conditions	59% (11, 81)	High
Pilishvili, 2018 [32]	US	Active Bacterial Core Surveillance (ABCs) IPD cases, adults ≥65 years	Case-control; controls identified as Medicare beneficiaries with no record of IPD or pneumonia	2015–2016	PCV13-type + 6C IPD	Gender, presence of chronic and immunocompromising conditions	47% (4, 71)	High
Prato, 2018 [33]	Italy	Inpatient and outpatient adults ≥65 years	Test-negative	2013–2015	PCV13-type pneumococcal pneumonia	Not reported	38% (−132, 89)	Low
**PPSV23**								
Andrews, 2012 [23]	UK	Adults ≥65 years	Indirect cohort	2003–2010	PPSV23-type IPD	Age and year of illness (implied by group matching)	48% (32, 60)	Medium
Chandler, 2022 [34]	US	Inpatient adults	Test-negative	2014–2017	PPSV23-type pneumococcal pneumonia	Age, diabetes, COPD, congestive heart failure, hyperlipidemia	2% (−50, 38)	High
Djennad, 2018 [24]	UK	Adults ≥65 years	Indirect cohort	2012–2016	PPSV23-type IPD	Age, clinical risk group, gender, year of notification, ethnicity	41% (23, 54)	High
Dominguez, 2005 [35]	Spain	Adults ≥65 years	Case-control; hospital controls	2001–2002	PPSV23-type IPD	Length of hospital stay, hospital period, COPD, use of corticosteroids, death	64% (31, 82)	Medium
Gutiérrez-Rodriguez, 2014 [25]	Spain	Adults ≥60 years	Indirect cohort	2008–2011	PPSV23-type IPD	Age, sex, year of symptom onset and presence of high-risk medical conditions	44% (19, 62)	High
Heo, 2022 [27]	South Korea	Hospitalized adults ≥65 years	Test-negative	2015–2017	PPSV23-type pneumococcal pneumonia	Age, sex, risk group based on underlying conditions, disease severity according to the CURB-65 score (confusion, urea level, respiratory rate, blood pressure, age ≥65), influenza vaccination status, and PCV13 vaccination status	6% (−74, 50)	High
Kim, 2019 [28]	South Korea	Adults ≥65 years	Case-control	2013–2015	PPSV23-type IPD	Chronic kidney disease, diabetes mellitus, smoking, and recent influenza vaccine exposure	42% (−2, 67)	High
					PPSV23-type non bacteremic pneumococcal pneumonia	Immunocompromised status, chronic heart disease, chronic pulmonary disease, chronic alcohol consumption, smoking, long-term care facility residence, and hospital center	−2% (−40, 26)	
Lawrence, 2020 [36]	England	Adults ≥16 years	Case-control	2013–2018	PPSV23-type pneumococcal pneumonia (PPSV23 non-PCV13 serotypes)	Age, sex, receipt of seasonal flu vaccination, and presence or absence of certain risk factors	46% (5, 69)	Medium
Perniciaro, 2021 [37]	Germany	Adults ≥60 years (German National Reference Center for Streptococci (GNRCS))	Indirect cohort	2018–2019	PPSV23-type IPD (PPSV23 non-PCV13 serotypes)	Age and gender	35% (−14, 65)	High
Rudnick, 2013 [21]	Canada	Adults ≥65 years	Indirect cohort	1995–2011	PPSV23-type IPD	Year of illness, gender	41% (21, 56)	Medium
Shimbashi, 2020 [38]	Japan	Adults ≥20 years	Indirect cohort	2013–2017	PPSV23-type IPD	Sex, age, prefecture, year, season, BMI group, underlying conditions, and smoking history with clustering by public health center	39.3% (−2.9, 64.2)	Medium
		Adults ≥65 years					39.4% (−6.1, 65.3)	
Su, 2021 [39]	Taiwan	Adults ≥75 years	Indirect cohort	2008–2016	PPSV23-type IPD		74% (39, 89)	High
			Screening method			Age and gender	43.4% (34.4, 51.2)	
Suzuki, 2017 [22]	Japan	Adults ≥65 years	Test-negative	2011–2014	PPSV23-type pneumococcal pneumonia	Study site, sex, age, underlying disorder, smoking status, pre-hospital antibiotic treatment, and year of hospital visit	34% (6, 53)	High

VE = vaccine effectiveness; IPD = invasive pneumococcal disease; CAP = community-acquired pneumonia; ^1^ Study funded by industry; ^2^ VE estimate presented as reported in study for overall estimate, all age groups, shortest time since vaccination (if stratified estimates provided); ^3^ Cochrane risk-of-bias tool used to assess quality of evidence.

## Data Availability

All data included for analysis were from the published literature.

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
