# Peer review of "Systematic Review and Meta-Analysis of the Efficacy and Effectiveness of Pneumococcal Vaccines in Adults"

_pathogens, 2023, doi:10.3390/pathogens12050732_

Round 1
Reviewer 1 Report
Meta analysis may be a source of substantial added value summarizing data of different setting and generalize them. I do think however we have enough data to draw any clear conclusion for pneumonia and in absence of enough RCTs conclusion will be always speculative. Also to generate conclusions for PCV15 or PCV20 based only on "similarity" with PCV13 and PPSV23 may be unrealistic.
Major comments:
1. Reason for review of PCV13 and PPSV23 effectiveness against clinical outcomes to help inform about the estimated effectiveness of PCV15 and PCV20 against clinical outcomes in adults seems to be very weak. The vaccines are licensed really on safety and immunogenicity data only, but they are not quite similar and rules for non-inferiority of more valent PCVs are less strict than rules originally valid in PCV7 period. So logically more valent vaccines are less immunogenic speaking e.g. about seven original serotypes and this is worsening with adding more serotypes.
2. Theoretical coverage of VT-serotypes is a possible marker, however it does not always fit to reality. Example is serotype 3, recently in many countries the second most frequent serotype, in spite of inclusion in PCV13. VE against serotype 3 in PCV20 may be even weaker, in some studies in PCV15 there is better immunogenicity with serotype 3 in some others not. So only practical proof of VE may show reality.
3. Selection of only 19 studies out of 5085 brings already high potential of bias, what maintain some very polar results particularly of VE against pneumonia.
4. Why the Prato study (2018) was included to such a stringent trials selection when even authors consider the study as weak?
Reviewer 2 Report
This is a useful update of previous analyses of the effectiveness of 13-valent pneumococcal conjugate vaccine and the 23-valent pneumococcal polysaccharide vaccine on IPD and pneumococcal pneumonia, but unfortunately has a very limited focus, namely vaccine-type disease. In light of all that we have learned regarding the dynamism of pneumococcal serotype epidemiology when under vaccine pressure, this scope limits its relevance, in my view, for public health decision-making, which should be concerned about vaccine impact on the overall incidence of pneumococcal disease. The phenomena of extensive serotype replacement, the limited or no effectiveness generally seen against serotype 3, as well as the question of differential efficacy of PCV13 and PPS23 against disease caused by the same serotypes, represent some of the more interesting and important open questions from PCVs to date that could be addressed here in an expanded set of analyses.
Specific concerns
1. First paragraph, introduction: some incidence figure for S. pneumoniae (which should be in italics) pneumonia should be provided in order to support the word "common" in line 1.
2. Through the manuscript, starting with the abstract, there is an implicit comparison of the efficacies of the two vaccines, as endpoints for both are referred to as “vaccine-type.” However, the vaccine types of course differ for the two vaccines, so these represent applies and oranges. Additional information that would help us better understand the relationship between antibody levels elicited and vaccine protection would be a systematic comparison of VE values by the two vaccines for disease caused only by the 13-serotypes. Is there clear evidence that VE against the 13 most successful serotypes (epidemiologically) for children is the same as VE against the other 11 serotypes in PCV23? Does PCV23 still appear to have slightly lower efficacy than PCV13 if only those types were examined? These authors could rigorously examine those open questions.
3. Introduction, line 54: Some comments about the continued prominence of a VT, serotype 3, despite vaccination, and the multiple impact studies casting serious doubt on the PCVs' true effectiveness, need to be offered, especially since the previous paragraph highlighted that it is the top serotype in adults. Don’t these papers collectively offer useful information on the relative VE against serotype 3 in particular by the two vaccines? Similarly, some explanation needs to be offered for lines 182-185: How does the serotype 3 incidence increase if there's a 53% VE?
4. Methods, line 98: Why is the primary and in fact, only endpoint, just VT disease? From both public health decision-making and clinical management perspectives, what happens with NVT IPD (and pneumonia) is just as important as IPD. Knowing what happens with NVT disease after adult vaccination matters also with regard to the validity of using the indirect cohort method to calculate efficacy. Why don’t the authors present estimates of VE against NVT disease too?
5. Section 2.4: There are many references to "adjustments" in VE, which the metanalyses rely on, but I didn't see any explanation as to on what basis they were made and whether those adjustments were made similarly by all authors.
6. Lines 198-203, Figure 1 and Figure 5: On what scientific basis can you include in the 23-valent metanalyses a VE value from one study that looks at only a subset of serotypes, while all other studies look at a broader range? The only justification would be if you show clear evidence that there is no difference in 23V PS vaccine effectiveness against the 11 “extra” types vs the PCV-13 types. Those kinds of analyses, as noted in #2, could be performed and would be valuable.
7. Discussion: The authors should explicitly compare their conclusions with those already reached by NIPH. In other words, what is new here with the addition of a few more years’ of studies, and what confirms what we already knew?
8. Lines 373-4: While these results are suggested to be useful for assessing the public health value of PCV15 and PCV20, the authors curiously avoid one of the more important (and currently inexplicable) geographical differences in the literature (“the elephant in the room”)--most countries do NOT report decreases in adult disease after widespread PCV13 use despite widespread PCV13 herd protection against VT, mostly due to increase in NVT disease. Furthermore, almost no countries followed the US CDC in recommending and funding adult vaccination with PCV13 due to doubts about incremental value. The authors should mention these issues, and discuss why they think adult vaccinations with 15 and 20 valent PCVs, which largely contain PCV13 types, would/should be viewed differently by policy makers (especially if only VT disease effectiveness data, and not overall pneumococcal disease effectiveness data, are being considered).
Round 2
Reviewer 1 Report
The manuscript was updated and reprocessed according the reviewers requests. Comments have been either accepted or at least explained.
Author Response
Dear Dr. Lee and Reviewer 1,
Thank you for taking the time to re-review our manuscript. We have made no further edits based on the additional comments from Reviewer 1.
Best regards,
Jennifer L Farrar
Reviewer 2 Report
The authors have adequately addressed a number of the concerns I had previously raised, but I still remain concerned about the sole focus on vaccine-type disease, rather than VT and NVT and overall disease, because it leaves the impression that VT disease is all that matters. The authors' argument for such a narrow focus is that effect on VT is "a vaccine-specific characteristic," which is true, but there are multiple examples in the literature (starting from Eskola et al NEJM AOM efficacy study (comparing Merck and Wyeth's PCV7 vaccines) to Henriques-Normark's Swedish PCV impact comparisons (of Pfizer's PCV13 and GSK's PCV10) to raise the question as to whether the effects on NVT (i.e., replacement) are also "vaccine-specific characteristics," not to mention potentially differential cross-protection of related serotypes. Accordingly, since the authors don't want to expand their analyses, at a minimum for acceptance I think the authors should acknowledge that this narrow focus on VT is a significant limitation of the analysis, since public health decision making on the value of these vaccines is not just about VT-efficacy or effectiveness measures.
